# The Role of Hematological Parameters in Predicting Fuhrman Grade and Tumor Stage in Renal Cell Carcinoma Patients Undergoing Nephrectomy

**DOI:** 10.3390/medicina55060287

**Published:** 2019-06-18

**Authors:** Erdem Kisa, Cem Yucel, Mehmet Zeynel Keskin, Mustafa Karabicak, Mehmet Yigit Yalcin, Ozgur Cakmak, Yusuf Ozlem Ilbey

**Affiliations:** 1Department of Urology, Tepecik Training and Research Hospital, 35180 Izmir, Turkey; 2Department of Urology, Batman Training and Research Hospital, 72070 Batman, Turkey

**Keywords:** red blood cell distribution width, neutrophil/lymphocyte ratio, renal cell carcinoma, Fuhrman grade

## Abstract

*Background and objective*: We investigated the ability of preoperative serum values of red blood cell distribution width (RDW), neutrophil lymphocyte ratio (NLR) and plateletcrit (PCT) to predict Fuhrman grades (FG) and tumor stages of renal cell carcinoma in patients who underwent radical nephrectomy. *Materials and methods*: Records of 283 patients that underwent radical or partial nephrectomy of renal masses at our clinic between January 2010 and April 2018, whose pathology results indicated renal cell carcinoma (RCC), and who had their FG and T1–4 N0M0 identified were retrospectively evaluated. The patients were divided into two groups based on their FG as low (I–II) and high (III–IV) and their T stages were similarly grouped as limited to kidney (pT1–pT2) and not limited to kidney (pT3–pT4). *Results*: Mean RDW, NLR, PCT cut-off values of the patients for FG and T stage were 15.65%, 3.54, 0.28% and 14.35%, 2.69, 0.28%, respectively. The RDW and NLR were determined to be statistically significant predictors of a pathologically high FG, whereas the PCT value was not a statistically significant predictor of high FG (*p* = 0.003, *p* = 0.006, *p* = 0.075, respectively). The relationship of RDW, NLR and PCT values with a limited to the kidney pathological T stage revealed statistically significant correlations for all three values. *Conclusions*: We determined that only RDW and NLR were markers predicting FG, while PCT had no prognostic value. On the other hand, all three of these values were associated with a limited to the kidney pathological T stage in patients who underwent nephrectomy due to renal masses and whose pathologies suggested RCC.

## 1. Introduction

Renal cancer comprises 2–3% of all cancers, and renal cell carcinoma constitutes the great majority of renal cancer cases. The incidence of renal cell carcinoma (RCC) demonstrated a yearly increase of 1.6% in a ten-year study spanning the years between 2002–2011 [1]. This increase primarily appeared due to the increased use of imaging methods for other reasons and the detected cases are usually incidentally localized renal masses [2]. Although imaging methods may provide insight about the malignant or benign nature of these renal masses prior to operation, they may not provide information regarding the aggressiveness and Fuhrman grade (FG) of the renal mass [3]. The FG along with age, tumor stage, tumor necrosis, performance status, and histological subtypes, is a significant pathological prognostic factor for RCC [4].

The red blood cell distribution width (RDW) is a parameter that measures variation in red blood cell size and is utilized in the differential diagnosis of certain types of anemia [5]. High RDW values have been shown to be an unfavorable prognostic factor for certain cancers and a negative predictor of overall and cancer-specific survival [6]. Moreover, this value was determined to have prognostic value for renal, prostate, and upper urinary tract urothelial tumors (UTUC) among urological cancers [7,8,9,10]. Another serological marker is the neutrophil-lymphocyte ratio (NLR), which can be tested preoperatively and can be used as a systemic marker of inflammation. It has been shown to predict the FG and prognosis in both clear cell RCC and non-clear cell RCC patients [11,12,13,14].

Platelets (PLT) are small anucleate cell fragments that play a crucial role in regulating hemostasis and managing vascular integrity. Mean platelet volume (MPV) is a value that represents the mean size of platelets found in the blood. MPV was shown to have prognostic value in cases of non-metastatic RCC [15]. Plateletcrit (PCT) count demonstrates the ratio of total platelets in the blood and is calculated by multiplying the number of platelets with MPV (PCT = PLT × MPV ÷ 10,000). This value has been investigated in relation to diseases such as acute cholecystitis in general surgery and preeclampsia in gynaecology [16,17]. In urology, it was investigated with regard to its relationship with varicocele, a vascular disease [18]. However, to our knowledge, its relationship with RCC has never been a subject of investigation before.

In this study, we aimed to determine the roles of RDW, NLR, PCT values in predicting high FG in patients that underwent partial or radical nephrectomy for renal masses and whose pathology results indicated RCC. We also aimed to investigate the relationship of these values with T stage, and to identify which of these values is a stronger predictor.

## 2. Materials and Methods

Records of 402 patients who underwent radical or partial nephrectomy for renal masses at our clinic between January 2010 and April 2018 were retrospectively evaluated. Approval of the institutional ethics committee was obtained (2019/1-6, 09.01.2019). Patients who had postoperative pathology results indicating RCC and had known T1–4 N0M0 and FG values were included in the study. Patients who had non-RCC malignant masses, masses that did not correspond to known RCC types, masses with benign pathology, chromophobe RCC, RCC patients with records lacking the FG, patients who had distant metastases at the time of diagnosis, patients who did not undergo surgical treatment, patients who had autoimmune diseases and other malignancies besides RCC, and patients who had incomplete data were excluded from the study (Figure 1). Demographic properties, operative data, and pathology results of patients were recorded. RDW, NLR, MPV, PLT, PCT (MPV × PLT ÷ 10,000) values of patients obtained 3–7 days preoperatively were acquired from their records. The normal reference ranges were specified as follows: RDW 11.6–17.2%, neutrophil 2.0–6.9 /uL, lymphocyte 0.6–3.4 /uL, PLT 140–400 /uL, MPV 6.0–11.0 fl, and PCT 0.17–0.7%. The patients were divided into two groups based on their FG: low (I–II) and high (III–IV) and their T stages were similarly grouped as limited to kidney (pT1–pT2) and not limited to kidney (pT3–pT4). The RDW, NLR, PCT values were compared in terms of their role in predicting pathologically low or high FG, their relationship with T stages limited and not limited to the kidney, as well as their sensitivity, specificity, and negative predictive values (NPV) and positive predictive values (PPV) percentages.

### Statistical Analysis

The median (min–max) was defined as descriptive statistic for the quantitative variables in the study, while frequency and percentages were defined for the categorical variables. In order to make discriminations within the FG and T stage groups, cut-off values of RDW, NLR, PCT were determined using Receiver Operating Characteristic (ROC) analysis. The sensitivity, specificity and area under curve (AUC) were determined for each value investigated in the study. The parameters were categorized based on the cut-off values calculated according to the Youden index, and the Pearson chi-square test was used to determine whether or not there were differences between the groups. For all tests, the probability of a type I error was considered α = 0.05. The statistical analyses in the study were performed using IBM SPSS V22.

## 3. Results

The demographic properties and pathology results of the patients are presented in Table 1. Based on pathological data, the low-FG group included 180 patients, the high-FG group had 103 patients, the group with a T stage limited to the kidney (pT1–pT2) had 193, and the group with a T stage not limited to the kidney (pT3–pT4) had 90 patients. The mean RDW, NLR, PCT cut-off values of the patients for FG and T stage were 15.65%, 3.54, 0.28% and 14.35%, 2.69, 0.28%, respectively. The FG and T stages, sensitivity, specificity, PPV, and NPV values associated with RDW, NLR and PCT values are presented in Table 2 and Table 3.

The RDW (>15.65) and NLR (>3.65) were determined to be statistically significant predictors of a pathologically high FG, whereas the PCT (>0.28) value was not a statistically significant predictor of high FG. The relationships of RDW (>14.35), NLR (>2.69) and PCT (>0.28) values with a pathological T stage limited to the kidney revealed statistically significant correlations for all three values. The relationships of RDW, NLR and PCT with pathological parameters are presented in Table 4.

## 4. Discussion

The relationship between cancer and systemic inflammatory process has been shown in previous studies [19,20,21]. Although the relationship between cancer biology and the systemic response of the body is complex, this relationship is founded on the hypothesis postulating that inflammatory cytokines (IL-1, IL-6, IL-8 and interferon-γ) released by the tumor trigger circulating acute phase reactants and hematological components including serum neutrophil counts [19]. These inflammatory cytokines also inhibit cytotoxic immune cells such as lymphocytes and T cells [20,21]. Various studies have demonstrated associations between inflammatory markers such as RDW, NLR, PCT that can be tested in the serum preoperatively, can be easily obtained, and have a prognostic value for certain diseases and cancers [6,7,8,9,10,11,12,13,15,16,17]. 

The relationship of FG and T stages of patients whose pathology results indicated RCC following radical or partial nephrectomy with their RDW and NLR values has been demonstrated in earlier studies [7,10,11,12,13]. According to our knowledge of the literature, our study is the first that investigates the relationship of PCT values with the FG and T stages in patients who underwent radical or partial nephrectomy due to renal masses indicative of RCC, and moreover, it is the first study that compares RDW, NLR, and PCT values in terms of predicting pathological FG, their relationships with T stages, and their sensitivity, specificity, PPV and NPV values. We determined that these three preoperative values were associated with the pathological T stages of patients, but that only RDW and NLR were able to predict a high FG.

The RDW value is a useful marker that has prognostic significance in various cancers [6,7,8,9,10]. Its role in predicting the prognosis of RCC and its relationship with the T stage were first demonstrated by Weng et al. They found higher serum RDW values in the presence of RCC when compared to the control group, and reported RDW to be correlated with a pathologically high FG and T stage [7]. Another study, which also investigated patients with RCC, found that high (>13.9%) and low (<13.9%) RDW values were linked to cancer-specific survival and that cancer-specific survival decreased as this value increased [8]. In our study, we found that the statistically determined RDW cut-off values predicted high FG and were correlated with T stage, and that these high FG and T stages demonstrated stronger NPV and PPV compared to NLR and PCT. 

The NLR value was shown to be correlated with the prognosis of certain solid cancers [22]. It was shown to be relevant in UTUC among urological cancers. In a study conducted by Dalpiaz et al., a high preoperative NLR value was reported to be associated with high cancer-specific and overall survival [23]. The relationship between NLR and RCC was investigated in a study conducted by Vier et al. They stated that the more aggressive subtypes of RCC (non-cystic, collecting duct type RCC) were associated with higher NLR values than the less aggressive subtypes (cystic and papillary type RCC). At the same time, they determined that as the NLR value increased from 2.65 to 4.77, the pathological FG of the RCC also increased. They suggested that this value could be useful in determining whether a renal biopsy should be obtained and influence the decision to administer surgical treatment [11]. Another study on RCC found that the NLR was associated with survival in metastatic RCC patients being treated with first line tyrosine kinase inhibitor [13]. Therefore, from determining the subtype and aggressiveness of RCC to predicting the response to tyrosine-kinase inhibitor in patients with metastatic RCC, the relationship of the NLR with RCC has been extensively investigated [11,12,13]. Paralleling the literature, our study also found that as the NLR value increased, the FG also increased, and that this value was related to the pathological T stage. However, we determined that when predicting FG, the PPV and NPV were lower than those of RDW.

PCT is a marker that has been used in various fields in the recent years and is calculated by multiplying the PLT count and the MPV value. The normal range for PCT is 0.22%–0.24% [24,25]. While the number of studies that demonstrate a connection of PCT with urological diseases is limited, there is a study that investigated its relationship with varicocele. In that study conducted by Polat et al., no statistical differences were determined in serum PCT values of patients with clinical varicocele compared to the control group [17]. We determined that the PCT cut-off value we had calculated statistically was associated with a high T stage, but statistically, it was not a predictive value foreseeing FG. 

In recent years, incidental small renal masses have been detected more frequently due to the increase in the availability of imaging methods such as ultrasonography and computed tomography [26,27]. Particularly for patients of advanced age and/or with comorbidities who are at a greater surgical risk, active surveillance is among treatment options as RCC-specific mortality is low. The sizes and yearly growth rates of the masses detected in the patients were followed-up using imaging methods [28,29]. One study found that a 1 cm increase in tumor size increased the possibility of the tumor being malignant by a rate of 16% and it being of a high grade by 25% [30]. In the case of an increase in the size of the renal mass during follow-up, the previously postponed treatment (surgery or ablation therapy) was initiated [28,29]. In addition to the size of the mass, its FG was shown to be connected to the prognosis of RCC subtypes [31]. However, currently, no tumor markers have been discovered that could provide insight about the FG of renal masses preoperatively. Moreover, the success of the percutaneous biopsy obtained from the renal mass at predicting FG has been limited [32]. However, we believe that serum markers such as RDW and NLR could be useful in making the decision to obtain a biopsy from patients under active surveillance due to renal masses as they could both provide information about whether the mass is a malignant or benign tumor and predict low or high FG if the mass is malignant. Furthermore, these markers could also be useful in making a decision between active follow-up and surgery in patients with unfavorable biopsy results.

The limitations of the study are its retrospective design, the involvement of multiple uropathologists in the evaluation of pathology results, and that the relationships of RDW, NLR and PCT values with overall and cancer-specific survival were not investigated.

## 5. Conclusions

The RDW, NLR and PCT parameters are inexpensive tests that can be obtained easily in a clinical setup. We determined that all three of these values were associated with the T stages of patients who underwent radical or partial nephrectomy due to renal masses and whose pathology results were suggestive of RCC. However, only RDW and NLR were determined as markers predicting FG, while the PCT had no effect in this regard. Although the current literature provides no suggestions regarding the use of these parameters, we believe that, following randomized controlled studies, they will be included in the nomograms of patients with renal masses who are planned to be monitored under active surveillance in the near future.

## Figures and Tables

**Figure 1 medicina-55-00287-f001:**
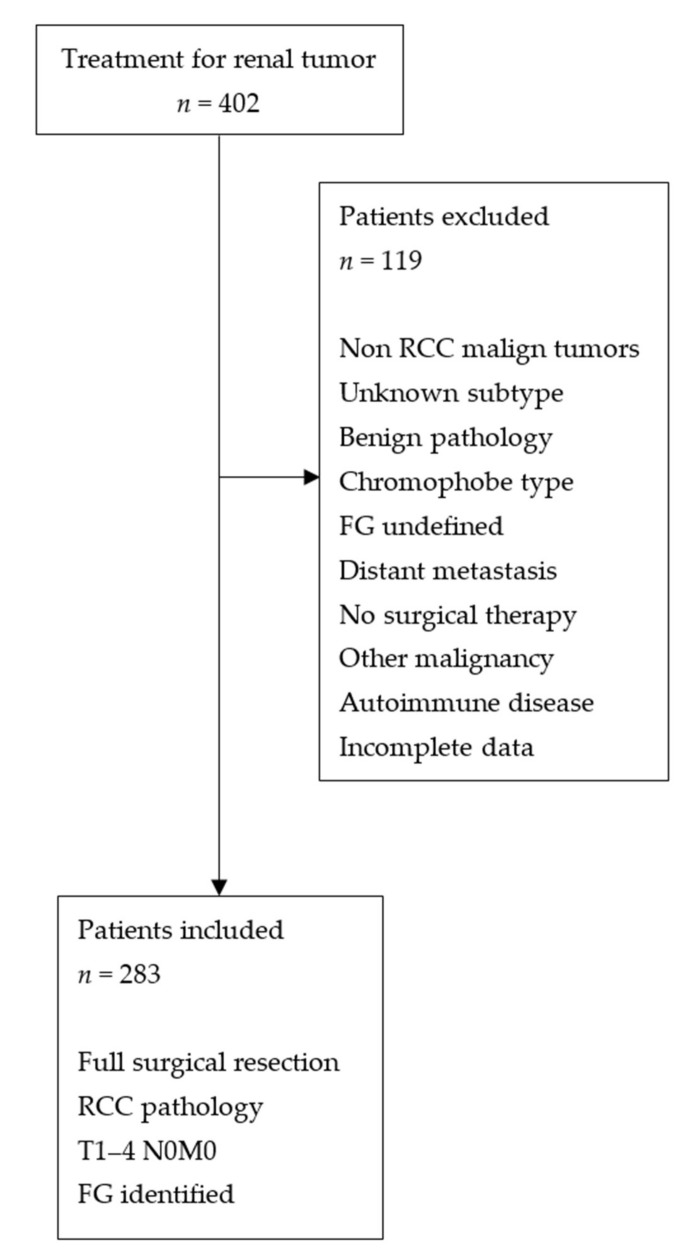
Flow chart of patients who met study inclusion/exclusion criteria. RCC = renal cell carcinoma, FG = Fuhrman grade.

**Table 1 medicina-55-00287-t001:** Patient characteristics and pathology outcomes.

Variable	*n* (%)
Number of patients	283
Median age, year (minimum–maximum)	61 (25–89)
Gender, *n* (%)	
Male	181 (63.9%)
Female	102 (36%)
Presentation type, *n* (%)	
Incidental	93 (32.8%)
Symptomatic	190 (67.1%)
Laterality, *n* (%)	
Right	128 (45.2%)
Left	155 (54.7%)
Median RDW, % (minimum–maximum)	14.4 (12.1–23.9)
Median neutrophil count, /uL (minimum–maximum)	5.3 (1.5–12.1)
Median lymphocyte count, /uL (minimum–maximum)	1.8 (0.2–4.6)
Median NLR, (minimum–maximum)	2.7 (0.82–14.13)
Median PCT, % (minimum–maximum)	0.232 (0.03–0.51)
Operation type, *n* (%)	
Radical Nephrectomy	221 (78%)
Partial Nephrectomy	62 (22%)
RCC subtype, *n* (%)	
Clear cell type	233 (82.3%)
Papillary type 1	31 (10.9%)
Papillary type 2	11 (3.8%)
Chromophobe type	8 (2.8%)
pT stage, *n* (%)	
Limited to the kidney (pT1–pT2)	193 (68.1%)
Non limited to the kidney (pT3–pT4)	90 (31.8%)
Fuhrman grade, *n* (%)	
Low grade (I–II)	180 (63.6%)
High grade (III–IV)	103 (36.3%)

RDW = red blood cell distribution width, NLR = neutrophil lymphocyte ratio, PCT = plateletcrit.

**Table 2 medicina-55-00287-t002:** Best cutoff values in which RDW, NLR, PCT can predict Fuhrman grade, sensitivity, specificity, PPV and NPV.

	AUC	95% CI		Cut-Off	The Maximum Youden’s Index	Sensitivity (%)	Specificity (%)	PPV	NPV	*p*
		Lower Bound	Upper Bound							
RDW	0.607	0.536	0.678	15.65	0.223	42.0	80.3	0.534	0.720	0.003
NLR	0.599	0.530	0.669	3.54	0.171	39.0	78.1	0.490	0.704	0.006
PCT	0.564	0.492	0.636	0.28	0.172	33.0	83.6	0.520	0.699	0.075

AUC = area under the curve, CI = confidence interval, PPV = positive predictive values, NPV = negative predictive values.

**Table 3 medicina-55-00287-t003:** Best cutoff values in which RDW, NLR, PCT can predict T stage, sensitivity, specificity PPV and NPV.

	AUC	95% CI		Cut-Off	The Maximum Youden’s Index	Sensitivity (%)	Specificity (%)	PPV	NPV	*p*
		Lower Bound	Upper Bound							
RDW	0.664	0.597	0.731	14.35	0.272	68.8	58.4	0.471	0.777	<0.001
NLR	0.621	0.552	0.691	2.69	0.213	64.5	56.8	0.509	0.695	0.001
PCT	0.573	0.500	0.647	0.28	0.155	32.3	83.2	0.446	0.748	0.045

**Table 4 medicina-55-00287-t004:** Association of RDW, NLR and PCT with pathological parameters.

	RDW <14.3, *n* = 140	RDW >14.3, *n* = 143	*p*
Pathologic Stage pT 1–2 pT 3–4	111 (58.4%) 29 (31.2%)	79 (41.6%) 64 (68.8%)	<0.001
	RDW <15.65, *n* = 205	RDW >15.65, *n* = 78	
Fuhrman grade I–II III–IV	148 (80.4%) 57 (57.6%)	36 (19.6%) 42 (42.4%)	<0.001
	NLR <2.6, *n* = 130	NLR >2.6, *n* = 153	*p*
Pathologic Stage pT 1–2 pT 3–4	99 (52.1%) 31 (33.3%)	91 (47.9%) 62 (66.7%)	0.003
	NLR <3.5, *n* = 202	NLR >3.5, *n* = 81	
Fuhrman grade I–II III–IV	142 (77.2%) 60 (60.6%)	42 (22.8%) 39 (39.4%)	0.003
	PCT <0.28, *n* = 221	PCT >0.28, *n* = 62	*p*
Pathologic Stage pT 1–2 pT 3–4	158 (83.2%) 63 (67.7%)	32 (16.8%) 30 (32.3%)	0.003

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
