# Peer review of "The Role of Hematological Parameters in Predicting Fuhrman Grade and Tumor Stage in Renal Cell Carcinoma Patients Undergoing Nephrectomy"

_1010-660X, 2019, doi:10.3390/medicina55060287_

Round 1

Reviewer 1 Report

In analyzing their results, the authors refer to a simplified 2-tiered Fuhrman grading score. What emerges from the study is that higher RDW and LNR are predictive of high FG of RCC. Moreover, RDW, NLR and PCT values are statistically significantly related to the limited (but not to the extended over the kidney) T stage. However, among the three hematological parameters, the highest NPV and PPV of high FG and T stage are for RDW cut-off values. For this reason, the scientific relevance of these findings needs to be enforced from a more extensive classification, in which these hematological values that are very simple to obtain could play a role.

As the authors themselves declare, benign masses and non-RCC malignancies were excluded from the analysis. Moreover, abnormal values for inflammatory markers like RDW, NLR and PCT can be also found in other diseases. Therefore, as resulting from the study, the three markers have not the power to provide information about the malignity of a renal mass that is under surveillance, neither to contribute to defining a surgical approach to an already biopsied renal cancer.

Author Response

Dear Reviewer,

We are thankful for your valuable input and comments about our study. We have engaged the services of professional editors and they have reviewed and edited the manuscript accordingly. The changes have been highlighted in bold.

Yours Sincerely,

Dr. Erdem Kisa

Fellow of European Board of Urology

University of Health Sciences Tepecik Training and Research Hospital, Department of Urology

Izmir/ TURKEY  

Reviewer 2 Report

Manuscript ID medicina-451798

Title: The role of hematological parameters in predicting Fuhrman grade and tumor stage in renal cell carcinoma patients undergoing nephrectomy

Review Comments:

More emphasis should be placed on the clinical value of the result obtained.

The different causal interpretations that the authors attribute to the results obtained should be discussed (what are the mechanisms through which RDW, NLR and PCT are affected?

Author Response

(The authors gave the same response as above.)

Reviewer 3 Report

The authors tried to investigate the association between preoperative serum values (red blood cell distribution width (RDW), neutrophil/lymphocyte ratio (NLR) and plateletcrit (PCT)) and the Fuhrman grades (FG) and tumor stages of renal cell carcinoma (RCC) in patients who underwent nephrectomy. The concept might be okay but some shortcomings need to be overcome.

 1. There was no statement of institutional review board approval in this human-subjects research. Obtaining informed consent is a basic ethical obligation and a legal requirement for researchers. The ethics perspective of this study should be involved.

2. The novelty of this study is relatively low, that is, RDW, NLR and MPV had been associated with grade or prognosis of RCC patients. Despite it should be informative to survey the association between those hematological parameters and RCC grade or stage in different racial/ethnic groups.

3. This study concluded that higher RDW and NLR associated with higher FG of RCC. And higher RDW, NLR and PCT associated with higher pathological stage of RCC. However it was uneasy to use those hematological parameters as predictive biomarkers and to help pathologists or surgeons precisely differential diagnosis for grade or stage. That is, it should not be a limitation to ignore their association with the overall and cancer-specific survival of recruited RCC population.

4. Some typographic and grammatical mistakes need to be revised more precisely before submission.

Author Response

(The authors gave the same response as above.)

Round 2

Reviewer 3 Report

1. Although this version of manuscript has been revised, the novelty is still relatively low. RDW, NLR and MPV had been reported to associate with poorer prognosis of RCC patients. This study concluded that higher RDW, NLR and MPV were associated with higher pathologic stages and RDW and NLR were associated with higher Fuhrman grade of postoperative RCC patients. Those findings were similar to most previous studies, but their prognostic values were ignored in this manuscript. It is necessary to show the association between RDW, NLR and MPV with the overall and cancer-specific survival among recruited RCC population. The prognostic meanings of those hematological parameters will be helpful for pathologists or surgeons to predict or improve the treatment effects of RCC patients.

2. In general, the experimental or analytic approaches of this study might be okay, but the novelty and clinical meanings of those three values for RCC patients were not clearly described in this manuscript.

Medicina EISSN 1010-660X Published by MDPI AG, Basel, Switzerland RSS E-Mail Table of Contents Alert
Back to Top